# Idiopathic Duodenal Hematoma: A Case Report and Literature Review

**DOI:** 10.3390/reports8020073

**Published:** 2025-05-19

**Authors:** Ebtesam Al-Najjar, Abdullah Esmail, Bayan Khasawneh, Saifudeen Abdelrahim, Maen Abdelrahim

**Affiliations:** 1Section of GI Oncology, Houston Methodist Neal Cancer Center, Houston Methodist Hospital, Houston, TX 77030, USAaesmail@houstonmethodist.org (A.E.);; 2Challenge Early College HS, Houston, TX 77081, USA; 3Faculty of Medicine, The University of Jordan, Amman 11942, Jordan

**Keywords:** idiopathic duodenal hematoma, conservative management, case report, non-traumatic hematoma

## Abstract

**Background:** Idiopathic duodenal hematoma is a rare clinical condition, typically associated with trauma, anticoagulation therapy, gastrointestinal procedures, or coagulopathies. We present a unique case of spontaneous duodenal hematoma in a patient without identifiable risk factors. **Case presentation:** We present the case of a 60-year-old Asian woman who presented to the emergency room (ER) with a 10-day history of progressive abdominal pain, early satiety, nausea, and vomiting. She had no history of trauma, anticoagulant use, or underlying predisposing conditions. On clinical evaluation, she was hemodynamically stable, and the initial laboratory results were unremarkable except for signs of dehydration and inflammation. A computed tomography (CT) scan revealed a heterogeneous lesion in the second portion of the duodenum, initially raising suspicion of a duodenal tumor. Further evaluation with magnetic resonance imaging (MRI) confirmed a duodenal hematoma with compression of the adjacent pancreas. **Management and Outcome:** The patient was managed conservatively with bowel rest, nasogastric decompression, intravenous (IV) fluid, and a proton pump inhibitor (PPI). Serial imaging demonstrated gradual hematoma resolution, with progressive improvement in her symptoms. She was discharged in stable condition and returned to normal activity after three weeks with complete hematoma resolution as seen on follow-up imaging. **Conclusions:** This case highlights the importance of considering spontaneous hematoma in the differential diagnosis of abdominal pain, even without risk factors. Early diagnosis and conservative treatment remain the mainstay of management and can lead to full recovery in uncomplicated cases.

## 1. Introduction

Duodenal hematoma is a rare condition, typically resulting from blunt abdominal trauma [1]. However, over 70% of duodenal hematoma cases in adults are due to non-traumatic conditions. These conditions include several pancreatic diseases, although the exact association remains unclear. Other contributing factors include coagulation disorders (such as anticoagulant therapy, hemophilia, Von Willebrand disease, and Henoch-Schönlein purpura), as well as other contributing factors include endoscopic procedures, connective tissue disorders, and idiopathic causes [2].

Clinical symptoms of duodenal hematoma can range from vague abdominal pain to acute abdomen, intestinal obstruction, and gastrointestinal (GI) bleeding. Diagnosis is typically confirmed using magnetic resonance imaging (MRI), computed tomography (CT), and upper gastrointestinal endoscopy (UGIE) [2].

Most cases of duodenal hematoma are resolved spontaneously or with the correction of coagulation abnormalities. However, in refractory cases, or when conditions such as malignancy, perforation, or intestinal tract obstruction are present, percutaneous drainage or surgery may be required [2].

However, cases of duodenal hematoma are rarely reported in the literature, particularly in the absence of identifiable risk factors. In this report, we present a case of an elderly woman with an idiopathic duodenal hematoma of unknown cause. She was successfully managed nonoperatively over three weeks and discharged without any specific complications.

## 2. Case Presentation

In this case report, we present a 60-year-old Asian woman with no significant medical history who arrived at the Emergency Room (ER) with a 10-day history of constant, progressive, and burning pain in the upper abdomen, rated 10/10 in intensity, accompanied by early satiety, nausea, and reflux. The patient had been taking nonsteroidal anti-inflammatory drugs (NSAIDS) and pantoprazole without relief. She then had worsening symptoms with the onset of clear vomiting and food mixed with fluids every 30 min to an hour associated with fatigue. Subsequently, the patient had an improvement in her symptoms. She had a CT scan the previous day, which revealed a large duodenal mass that prompted a concern for malignancy and hence further evaluation.

Upon examination in the ER, she was alert and oriented, with a blood pressure (BP) of 153/90 mmHg, a heart rate (HR) of 94 beats/min, oxygen saturation of 95% on room air, and a respiratory rate (RR) of 16/min. Her past medical history was unremarkable. Physical exam revealed a flat abdomen with severe tenderness in the epigastric region. The basic laboratory test results, including complete blood count (CBC), electrolytes, prothrombin time (PT), partial thromboplastin time (PTT), international normalized ratio (INR), C-reactive protein (CRP), pancreatic enzymes, and liver chemistries reflected a state of dehydration and systemic inflammation but were otherwise unremarkable. Antinuclear antibody (ANA) screening was negative.

The patient was managed conservatively with nil per oral (NPO), insertion of a nasogastric (NG) tube, and initiation of intravenous (IV) fluids, a proton pump inhibitor (PPI), and pain medication.

A CT scan of the abdomen revealed a heterogeneous lesion measuring 6.1 cm × 8.8 cm × 8.7 cm in the second portion of the duodenum with mild heterogeneous enhancement (Figure 1A). Given the imaging features, a duodenal malignancy was initially suspected. The attending physician decided to pursue further investigations to rule out malignancy, and the patient was admitted for a comprehensive evaluation to establish a definitive diagnosis. The following day, an MRI cholangiogram with contrast was performed, revealing a lesion measuring 7.8 cm × 5.8 cm, in the second and third parts of the duodenum. The lesion exhibited heterogeneity with a lack of enhancement, indicative of a duodenal hematoma (Figure 1B). The hematoma affected the duodenum and the head of the pancreas, likely causing extrinsic compression with mild bile duct dilation.

The patient was diagnosed with a duodenal hematoma, and surgical intervention was discussed with the surgical team. However, due to the patient’s stable condition, the surgeon recommended continuing with medical management. By the third day, her symptoms, including vomiting and abdominal pain, had improved. It was recommended to continue NPO with an NG tube. An esophagogastroduodenoscopy (EGD) was performed, revealing several non-bleeding gastric ulcers in the gastric antrum, likely secondary to NG tube trauma (Figure 2A). Additionally, severe acquired extrinsic stenosis was observed in the second portion of the duodenum (Figure 2B).

A follow-up MRI revealed mild thickening and mucosal enhancement in the second and third portions of the duodenum, along with a significant decrease in the size of the adjacent duodenal hematoma. However, the patient continued to experience symptoms of gastric outlet obstruction. In response, nasojejunal (NJ) tube feeding was initiated alongside NG tube suction with a clamp trail. Subsequently, there was a decrease in suction output with symptom improvement.

Over the following days, the NG tube suction output remained significant (1–1.5 L daily), and the liver function tests (LFTs) showed a slow uptrend with mixed hepatocellular and cholestatic pattern, likely due to extrinsic compression on the distal common bile duct from the duodenal hematoma. NJ feeding continued, with close monitoring of NG suction output. The patient tolerated the NJ feeds well, and the LFT showed slight improvement. A repeated MRI revealed a reduction in the hematoma size (from 7.8 cm × 5.8 cm to 6.2 cm × 2.5 cm as shown in Figure 3), along with decreased duodenal wall thickness and hyperenhancement. A follow-up EGD revealed improved non-bleeding gastric ulcer and successful passage of the scope through the second portion of the stenosed duodenum.

The NG tube was clamped again for 13 h with reduced suction output and no vomiting. The patient continued to have bowel movements and passed flatus. Following this successful trial, NJ tube feeds were continued until she was able to tolerate adequate oral intake. Eventually, both tubes were removed, and the patient tolerated oral liquids and soft foods well during her recovery.

The patient showed significant improvement with conservative management and was discharged after three weeks, with a scheduled follow-up in outpatient care. A surveillance scan with IV contrast indicated complete resolution of the hematoma (Figure 4). The patient remained asymptomatic and was able to resume normal activities. A timeline detailing the patient’s diagnosis and treatment history is provided in (Figure 5).

## 3. Discussion

The duodenum is the widest portion of the small bowel, and it lacks a mesentery. It is divided into four parts: first, second, third, and fourth. The first part is intraperitoneal, while the remaining three are retroperitoneal. The retroperitoneal attachment and lack of mesentery, particularly in the third portion near the spine, make it susceptible to blunt abdominal trauma [3].

A duodenal hematoma was reported first by McLauchlan from an autopsy in 1838. The hematoma was caused by a pseudoaneurysm, where the cause was unclear but presumed to be trauma-related [4,5]. The first documented non-traumatic case of intramural duodenal hematoma was reported by Sutherland in 1904, in a child with Henoch-Schönlein purpura. Later, in 1908, Von Khautz diagnosed a similar condition in a patient with hemophilia [6].

Forty articles reporting 103 cases of spontaneous intramural small bowel hematoma (SISBH) were found in a review of the literature from the MEDLINE database conducted over 30 years ago. These included 39 patients with jejunal hematoma, 25 with ileal hematoma, and 18 with duodenal hematoma. Diffuse segments of the small intestine were involved in three cases, while the others were unspecified. Additionally, a retrospective cohort study published in 2019 by Kang et al., included 37 patients, with duodenal hematoma found in 4 patients, jejunal in 16 patients, and ileal in 17 patients [7]. This case supports the previous literature by highlighting a rare idiopathic duodenal hematoma.

The etiology of duodenal hematoma is not fully understood; however, the most common clinical causes involve blunt abdominal trauma [8]. However, conditions such as anticoagulant therapy, pancreatitis, bleeding disorders, malignancy, vasculitis, and upper endoscopy have also been associated with non-traumatic cases [9]. According to research conducted in Switzerland, the estimated incidence of SISBH related to anticoagulation is approximately 1 in 2500 anticoagulated patients per year [10]. Our patient is regarded as an uncommon idiopathic case occurring without trauma or identifiable risk factors.

The clinical presentation can vary depending on the location of the hematoma, with symptoms of either high or low GI obstruction often being predominant. The main symptom is abdominal pain, which typically persists for several days before symptoms of GI obstruction appear. The progression of symptoms and potential complications may include blood loss, bowel gangrene, intussusception, or obstruction. Acute rupture of the hematoma can lead to hemodynamic instability [11].

In cases of SISBH, abdominal pain and vomiting are highly prevalent, occurring in 84.6% to 100% of cases. However, hematemesis and fever are less common, presenting in only 15.3% to 23.1% of cases [11]. This is consistent with the clinical presentation of our patient who had abdominal pain and non-bloody vomiting.

Given that the mortality rate may reach 30%, early diagnosis is essential. Accurate diagnosis is challenging for several reasons, including its retroperitoneal location, which limits the ability to detect it during clinical examination. In addition, the incidence is low, as only 5% of abdominal trauma cases lead to duodenal injury, and spontaneous cases are even rarer [12].

Regarding the evolution of the diagnostic techniques for duodenal hematoma, Felson and Levin, in their 1954 report of four cases, highlighted distinctive X-ray findings in duodenal intramural hematoma cases. They also highlighted Liverud’s 1948 article as the first detailed description of radiographic features in this context. Felson and Levin emphasized that demonstrating an intramural mass with an overlaying coil-spring mucosal pattern is pathognomonic for a duodenal intramural hematoma [13]. Barium X-ray was widely used in the diagnosis of duodenal hematoma until the 1980s, due to the emergence of more advanced motilities such as CT and ultrasound imaging. These methods are less invasive, faster, and easier to perform, providing a more comprehensive evaluation of free fluid in the abdominal cavity and retroperitoneum. The combination of CT and ultrasound has demonstrated 100% diagnostic accuracy, as shown by Polat and colleagues in 2003 [10]. In a case reported by Al-Mwald et al., it was mentioned that a CT scan is considered the most specific and sensitive in the diagnosis of hematoma [14]. MRI or EGD have recently been utilized to improve visualization of the lesion’s location, size, and relationship to surrounding structures, as well as to detect early signs such as bile duct dilation or abnormalities in the pancreatic tissue [3,15]. In our case, a CT scan was initially performed as the imaging modality of choice for a patient presenting with abdominal pain, due to its shorter acquisition time. However, the findings presented a suspicion of malignancy, prompting the need for further evaluation. An MRI, which revealed a heterogeneous lesion and lack of enhancement, proved to be more effective in the diagnosis of hematoma.

Surgical management was commonly performed in the past. However, it is now only required in cases of intra-abdominal complications, like uncontrolled bleeding or peritonitis [16]. Regardless of whether the hematoma is traumatic or non-traumatic in origin, several retrospective studies have exhibited the effectiveness of conservative management [16]. According to a literature review of the MEDLINE database, among 103 patients with SISBH, 18 required surgical intervention due to complications such as peritonitis, intestinal obstruction, or necrotizing pancreatitis. In contrast, a cohort study by Kang et al., reported that all 37 patients with SISBH recovered following conservative management without the need for surgery [7]. Typically, duodenal hematomas, even those causing obstruction, are managed conservatively. This approach includes digestive rest, NG decompression, blood transfusions, and correcting any coagulation abnormalities. Surgery is considered if there is no improvement after two weeks of treatment, if there is a suspicion of neoplasia, or if major complications arise [6]. In this case, the patient successfully recovered through conservative management.

## 4. Conclusions

Our patient presented with abdominal pain and non-bloody vomiting and improved with supportive treatment. This study highlights the importance of considering duodenal hematoma among the differential diagnosis in patients presenting with abdominal pain even with the absence of trauma or relevant risk factors and the benefit of early diagnosis and supportive treatment in achieving a successful recovery.

## Figures and Tables

**Figure 1 reports-08-00073-f001:**
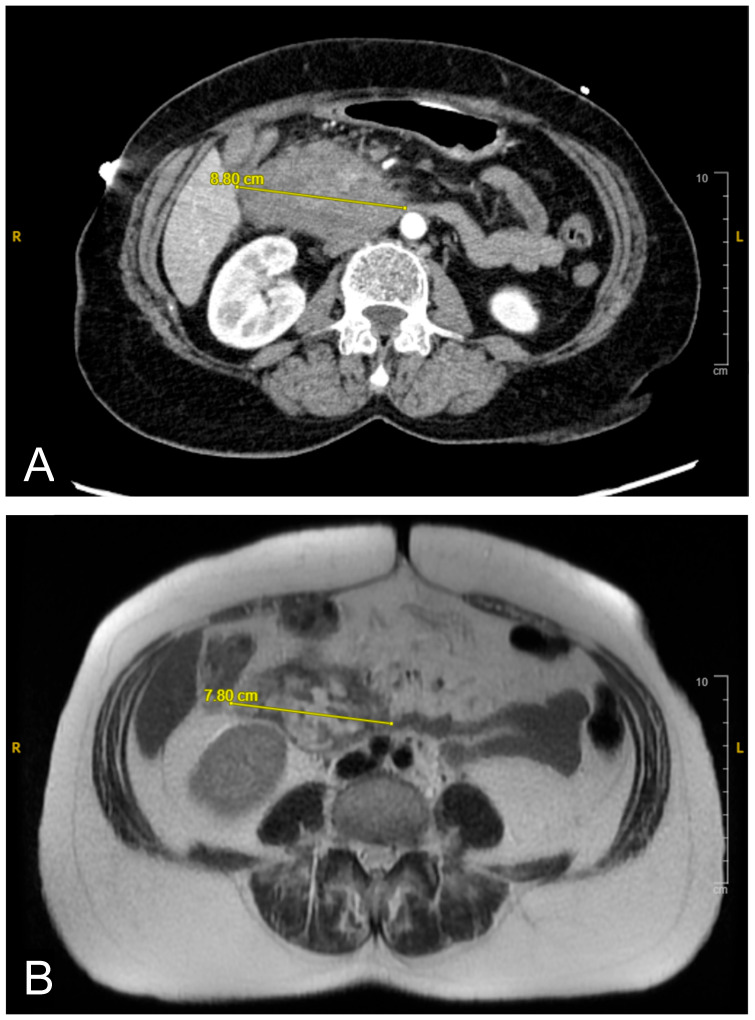
(**A**): Initial contrast-enhanced abdominal CT scan revealing duodenal hematoma measuring 8.8 cm in its widest diameter. (**B**): Follow-up MRI cholangiogram showing interval regression in size to 7.8 cm. Image quality was limited by motion artifact.

**Figure 2 reports-08-00073-f002:**
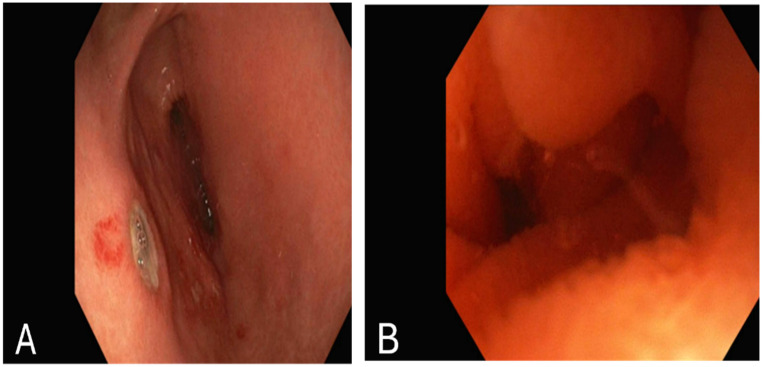
Esophagogastroduodenoscopy showing: (**A**): several non-bleeding gastric ulcers in the gastric antrum. (**B**): stenosis in the second part of the duodenum.

**Figure 3 reports-08-00073-f003:**
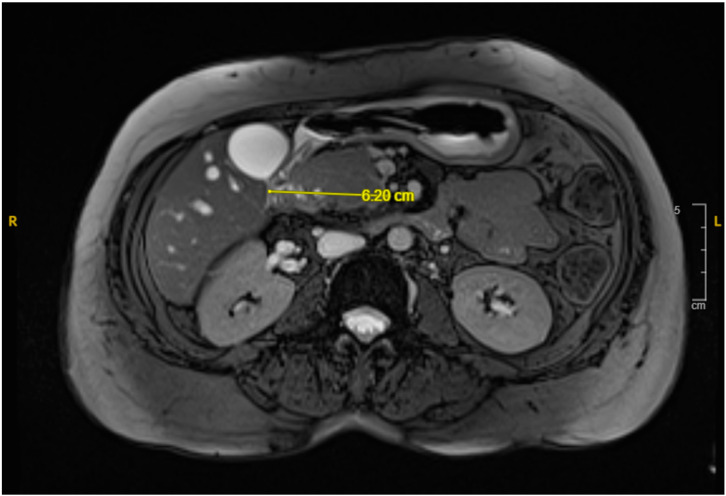
MRI cholangiogram revealing interval decrease in size of the duodenal hematoma to 6.2 cm in its widest diameter.

**Figure 4 reports-08-00073-f004:**
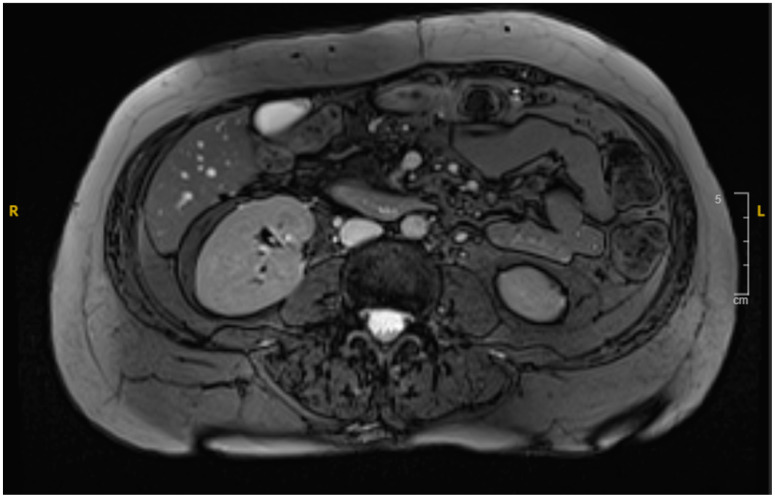
MRI revealing resolution of the duodenal hematoma.

**Figure 5 reports-08-00073-f005:**
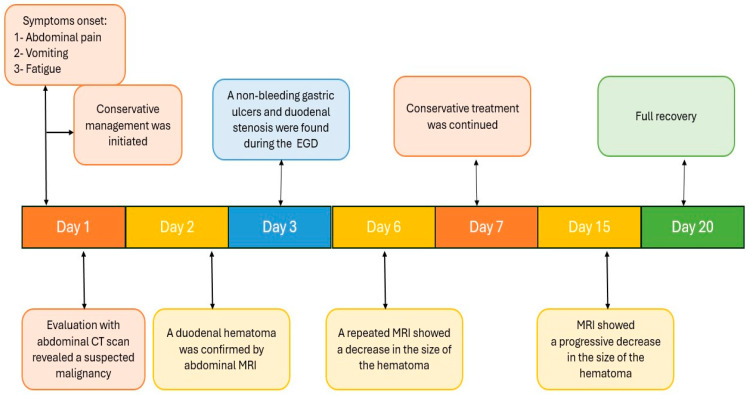
Timeline illustration revealing the patient’s diagnosis and treatment details.

## Data Availability

The data of this case report are available upon request from the corresponding author, Maen Abdelrahim (mabdelrahim@houstonmethodist.org).

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
