# Peer review of "Idiopathic Duodenal Hematoma: A Case Report and Literature Review"

_reports, 2025, doi:10.3390/reports8020073_

Round 1

Reviewer 1 Report

Comments and Suggestions for Authors

Interesting article, however, this article needs, few clarification.

  1. The author did not give valid reasons why CT could not give the diagnosis, but MRI gave the diagnosis.
  2. Entire case report is based on MRI image. But MRI image provided has “motion artifact degrades”. No clear-cut hematoma is seen. Patients had undergone several MRI, but good quality image is lacking.
  3. No details about work up for the cause of hematoma like coagulation profile, vasculitis or local cause like diverticula/pancreatitis have been provided.
  4. This patient was symptomatic before severe pain abdomen occurred, how you correlate this chronology. Has any attempt been made to find out the reason for burning pain for several days.
  5.  

Author Response

Title: Idiopathic duodenal hematoma management conservatively: A case report and literature review 

Journal: MDPI 

Manuscript ID: Reports-3576087 

Received: 03/24/2025 

Dear Drs.  Reviewer, Celia and Liu,

We would like to thank you the learned reviewers and editor for consideration of our manuscript for publication and thoroughly appreciate the time taken to provide us with valuable comments to improve the readability and value for our contribution to literature. We have provided responses to all the comments below. 

  1. Comments and Suggestions for Authors

Interesting article, however, this article needs few clarifications. The author did not give valid reasons why CT could not give the diagnosis, but MRI gave the diagnosis. 

Response: Thank you very much for this valid point. We have added an explanation in case presentation section. CT was inconclusive and raised suspicion of malignancy, whereas MRI, with its superior soft-tissue resolution, clearly identified features consistent of heterogeneity, lack of enhancement), leading to the correct diagnosis 

  1. Entire case report is based on MRI image. But MRI image provided has “motion artifact degrades”. No clear-cut hematoma is seen. Patients had undergone several MRI, but good quality image is lacking.

Response: Thank you for your valuable feedback. We added a new MRI and indicated the site and dimensions of the duodenal hematoma.

  1. No details about work up for the cause of hematoma like coagulation profile, vasculitis or local cause like diverticula/pancreatitis have been provided.

Response: We appreciate your review. We have added the details about work up. 

  1. This patient was symptomatic before severe pain abdomen occurred, how you correlate this chronology. Has any attempt been made to find out the reason for burning pain for several days.

Response: Thank you for your comment. We have included the medications the patient was taking before presenting to the Emergency Room 

Thank you, 

The team 

Reviewer 2 Report

Comments and Suggestions for Authors

The authors reported a case of idiopathic duodenal hematoma which is an uncommon condition and perhaps under-recognised and diagnosed. Therefore, there is some merit to the case. However, there are many issues with the case presentation.

Major issues

The is not well written and presented. I am not certain what case is this. Is it a duodenal hematoma (intramural) or paraduodenal hematoma? These are distinct entities. From the case report presentation, the authors mentioned that this case is a paraduodenal hematoma but the rest of the manuscript including the title is about duodenal hematoma.

The description of the case progress can be summarised. 

The MRI image is distorted (stretched sideway) and one cannot appreciate where the pathology is; Please adjust and also place marker to indicate the pathology. Otherwise use a CT image instead which can be appreciated better.

Factual error (Introduction section); The first case of non traumatic intramural hematoma was not reported or described by Henoch-Schonlein. It was reported by McLauchlan in 1838.

In the discussion section, the authors divided the duodenum into superior, descending, horizontal and ascending, a categorization that is more commonly used will be the first, second, third and fourth part of the duodenum. Perhaps, can include this in parenthesis. Otherwise, leaving it as it is, is acceptable.

Inappropriate use of abbreviations in the abstracts and keywords section.

Some of the references are very hold and can be updated.

Conclusion is poor and too short, especially the second sentence. This statement 'Understanding this condition is crucial for establishing timely diagnosis and ruling out other potential causes' is true for any condition. But what is so special about this particular case.

In the ethical section; why was an institutional board review required (unless is compulsory for the institution) and a consent not required (which is a general requirement for a case report/series).

Comments on the Quality of English Language

Needs a lot of improvement. 

Author Response

Title: Idiopathic duodenal hematoma management conservatively: A case report and literature review 

Journal: MDPI 

Manuscript ID: Reports-3576087 

Received: 03/24/2025 

Dear Drs. Reviewer,  Celia and Liu,

We would like to thank you the learned reviewers and editor for consideration of our manuscript for publication and thoroughly appreciate the time taken to provide us with valuable comments to improve the readability and value for our contribution to literature. We have provided responses to all the comments below. 

  1. Comments and Suggestions for Authors

  1. This is not well written and presented. I am not certain what case is this. Is it a duodenal hematoma (intramural) or paraduodenal hematoma? These are distinct entities. From the case report presentation, the authors mentioned that this case is a paraduodenal hematoma but the rest of the manuscript including the title is about duodenal hematoma.

Response: Thank you for your review. We have standardized the diagnosis as duodenal hematoma throughout the entire manuscript. 

  1. The description of the case progress can be summarised. 

Response: Thank you for the feedback. We have summarized the case progress. 

  1. The MRI image is distorted (stretched sideway) and one cannot appreciate where the pathology is; Please adjust and also place marker to indicate the pathology. Otherwise use a CT image instead which can be appreciated better.

Response: Thank you for your valuable input. The previous image was limited by an artifact but now we have added a new MRI showing the dimensions of the duodenal hematoma.

  1. Factual error (Introduction section); The first case of non traumatic intramural hematoma was not reported or described by Henoch-Schonlein. It was reported by McLauchlan in 1838.

Response: Thank you for bringing the factual error in the Introduction section to our attention. The first case of traumatic intramural hematoma was reported by McLauchlan in 1838, while the first documented non-traumatic case of intramural duodenal hematoma was described by Sutherland. We have revised the text accordingly, provided clarification, and we moved to the discussion section, where it fits more appropriately.  

  1. In the discussion section, the authors divided the duodenum into superior, descending, horizontal and ascending, a categorization that is more commonly used will be the first, second, third and fourth part of the duodenum. Perhaps, can include this in parenthesis. Otherwise, leaving it as it is, is acceptable.

Response: I appreciate your comment. We have paraphrased it. 

  1. Inappropriate use of abbreviations in the abstracts and keywords section.

Response: Thank you for your comment. We have fixed it. 

  1. Some of the references are very old and can be updated.

Response: Thank you for bringing this up. We have removed all the old references.  

  1. Conclusion is poor and too short, especially the second sentence. This statement 'Understanding this condition is crucial for establishing timely diagnosis and ruling out other potential causes' is true for any condition. But what is so special about this particular case.

Response: Thank you for your feedback. We have revised the conclusion to specifically highlight what makes this case unique. 

  1. In the ethical section; why was an institutional board review required (unless is compulsory for the institution) and a consent not required (which is a general requirement for a case report/series).

Response: Thank you for your feedback. Our institution mandates IRB review for all case reports to ensure compliance with ethical standards and institutional policies, even when patient data is fully anonymized. The IRB review was conducted to confirm that the case report adhered to ethical guidelines, including the protection of patient privacy and confidentiality, as per our institutional requirements. Although the IRB determined that formal patient consent was not strictly required due to the retrospective nature of the study and the complete anonymization of patient data (in compliance with the Declaration of Helsinki and applicable privacy regulations, such as HIPAA or equivalent local standards), we proactively obtained written informed consent from the patient to avoid any potential confusion and to uphold the highest ethical standards. The consent process ensured the patient was fully informed about the use of their de-identified medical data for this case report.

  1. Comments on the Quality of English Language: Needs a lot of improvement. 

Response: We appreciate your comment. We have reviewed the manuscript, corrected all errors, and improved the grammar to enhance readability. 

Thank you, 

The team 

Round 2

Reviewer 1 Report

Comments and Suggestions for Authors

Case report based MRI images- You have provided only initial MRI

Please provide repeat MRI (last image showing complete resolution of hematoma, also repeat Endoscopy images.

Figure 3 is irrelevant

Please mark the changes in red font, so that repetition is avoided

Comments on the Quality of English Language

fine

Author Response

Dear Reviewer,

Thank you for your valuable feedback on our case report. We have carefully considered each of your comments and made the necessary revisions to address them. Below, we respond to each point individually, outlining the changes made to improve the manuscript.

  1.  You have provided only initial MRI. Please provide repeat MRI (last image showing complete resolution of hematoma, also repeat Endoscopy images.
    • Response: We appreciate your suggestion to include follow-up imaging to demonstrate the resolution of the hematoma. We have added the repeat MRI image, which shows complete resolution of the hematoma, as a new figure.
  2. Comment: Figure 3 is irrelevant.
    • Response: Thank you for pointing this out. Figure 3 is the timeline illustration revealing the patient’s diagnosis and treatment details and we would like to clarify its significance to the case report. Figure 3 serves as a concise visual summary of the patient’s clinical course, integrating key diagnostic milestones (e.g., initial MRI and endoscopy findings) with treatment interventions and follow-up outcomes. This timeline is particularly valuable for readers, as it contextualizes the progression of the case, highlights the temporal relationship between interventions and resolution, and enhances the report’s educational value. For example, the timeline clearly delineates the sequence of diagnostic imaging, therapeutic decisions, and the eventual resolution of the hematoma, which may not be as easily conveyed through text alone. 
  3. Comment: Please mark the changes in red font, so that repetition is avoided.
    • Response: we have marked all changes in the revised manuscript with tracking tool , as requested. This includes updates to the text describing the new MRI and endoscopy images, removal of Figure 3, and any associated revisions to the figure legends and text. The use of this tool highlights the modifications for easy identification and confirms that no redundant content remains in the manuscript.

We believe these revisions fully address your comments and enhance the clarity and quality of the case report. Please let us know if there are any additional changes or clarifications required.

Sincerely,
The team